# Copper Utilization, Regulation, and Acquisition by *Aspergillus fumigatus*

**DOI:** 10.3390/ijms20081980

**Published:** 2019-04-23

**Authors:** Nicholas Raffa, Nir Osherov, Nancy P. Keller

**Affiliations:** 1Department of Medical Microbiology and Immunology, University of Wisconsin-Madison, Madison, WI 53706, USA; nraffa@wisc.edu; 2Department of Clinical Microbiology and Immunology, Sackler School of Medicine, Tel-Aviv University, Ramat-Aviv 69978, Tel-Aviv, Israel; nosherov@tauex.tau.ac.il; 3Department of Bacteriology, University of Wisconsin-Madison, Madison, WI 53706, USA

**Keywords:** copper homeostasis, *Aspergillus fumigatus*, secondary metabolites

## Abstract

Copper is an essential micronutrient for the opportunistic human pathogen, *Aspergillus fumigatus*. Maintaining copper homeostasis is critical for survival and pathogenesis. Copper-responsive transcription factors, AceA and MacA, coordinate a complex network responsible for responding to copper in the environment and determining which response is necessary to maintain homeostasis. For example, *A. fumigatus* uses copper exporters to mitigate the toxic effects of copper while simultaneously encoding copper importers and small molecules to ensure proper supply of the metal for copper-dependent processes such a nitrogen acquisition and respiration. Small molecules called isocyanides recently found to be produced by *A. fumigatus* may bind copper and partake in copper homeostasis similarly to isocyanide copper chelators in bacteria. Considering that the host uses copper as a microbial toxin and copper availability fluctuates in various environmental niches, understanding how *A. fumigatus* maintains copper homeostasis will give insights into mechanisms that facilitate the development of invasive aspergillosis and its survival in nature.

## 1. Introduction

All living organisms have evolved to maintain metal homeostasis as many metal ions are essential for certain biological processes, but high concentrations of these very same ions are toxic. Transition elements such as iron, copper, nickel, and zinc are required as cofactors for processes such as electron transfer or for maintaining protein structure. Estimates show that approximately 50% of all proteins require some metal cofactor to function [1,2]. However, these elements have a darker side. If microbes are not able to carefully regulate homeostasis of the metal, it can quickly become toxic, damaging macromolecules, affecting homeostasis of other metals, and ultimately killing the cell. 

Metal homeostasis is also critical in host/pathogen interactions. The best studied transition metal in that regard is iron [3,4]. However, several recent studies have highlighted the role of copper in these interactions. Copper is a transitional metal that has two biologically relevant oxidation states, Cu^+^ and the more soluble Cu^+2^. Of all the cellular transition metals, copper establishes the most stable ligand complexes according to the Irving–Williams series [5], resulting in a high tendency to compete with and displace other metal cofactors [6]. Copper also participates in the generation of reactive oxygen species (ROS) through Fenton chemistry, a process host cells use to kill pathogenic microbes [7]. In turn, microbes have evolved several mechanisms to counteract host copper-mounted attacks [8]. 

Here we seek to provide a comprehensive synopsis of the current state of knowledge of how the opportunistic human pathogen *Aspergillus fumigatus* regulates copper homeostasis and the role of copper in host–fungal interactions. This pathogen causes invasive aspergillosis (IA) in immunocompromised hosts, with mortality rates as high as 90% [9]. The study of copper homeostasis in fungi was initiated in the model yeast *Saccharomyces cerevisiae* [10] and *Schizosaccharomyces pombe* [11]. Subsequent work has mainly focused on copper regulation in the pathogenic fungi *Cryptococcus neoformans*, *Candida albicans*, and recently *A. fumigatus* [11,12]. We compare and contrast the system in these fungi and provide a view of the potential small molecule copper biology of *A. fumigatus*.

## 2. Copper Metabolic Processes

All living organisms require copper for several biological processes ranging from cofactor requirements for enzyme activity to transcription factor functionality. Delivery of the copper cofactor to specific proteins is a function of copper chaperones, although compared to the number of copper-containing proteins there have been only a few chaperones demonstrated to be responsible for delivery of the cofactor. 

### 2.1. Enzymes Using Copper as A Cofactor

Possibly the most important role of copper is its requirement for heme–copper oxidases such as cytochrome c oxidase [13]. Cytochrome c oxidase, CycA in *A. fumigatus*, is a ubiquitous protein in eukaryotes that is essential for facilitating energy generation by catalyzing the final step in the electron transport chain [14]. Inhibition of the cytochrome c oxidase complex assembly in mammalian cells or loss of Rcf1 and Rcf2, yeast proteins supporting the cytochrome c oxidase complex, results in defects in aerobic respiration [15] indicating the importance of the complex in energy generation [14]. Superoxide, a toxic ROS by-product of respiration [16,17] is detoxified by copper-containing superoxide dismutases (SODs). Fungi use these copper SODs to convert the superoxide into hydrogen peroxide, which is subsequently converted into water and oxygen by an iron-dependent catalase [18,19]. 

Copper is also essential for the uptake and utilization of other nutrients. The metabolism of a variety of nitrogen sources requires a copper-dependent process. Primary amines, a source of nitrogen, can only be utilized as a nitrogen source by an amine oxidase, a protein requiring copper as a cofactor. The *A. fumigatus* genome contains five putative copper-binding amine oxidases, none of which have been thoroughly investigated. Some nitrite reductases utilize copper as a cofactor and reduce nitrate to nitrite, another step in utilizing nitrogen for amino acid biosynthesis [20]. *A. fumigatus* contains a known nitrite reductase, NiiA, that has not been shown to bind copper, but contains another putative nitrite reductase (Table 1) that contains a copper-binding motif. Nitrite reductase also plays an important role in cell communication and development as nitrous oxide is a major signaling molecule described in bacteria, plants, fungi, and mammals [21,22]. Reductive iron uptake, the process by which *A. fumigatus* reduces ferric iron (Fe^+3^) to the more soluble ferrous iron (Fe^+2^) is most likely catalyzed by the copper-utilizing ferroxidase, FetC [23]. Cu^+^ oxidation is coupled to the reduction of iron, which is then transported into the cell via the iron permease FtrA [23]. The exact mechanism by which copper and iron are involved in iron uptake has not been experimentally determined in *A. fumigatus*. 

Copper is required for the biosynthesis of some secondary metabolites, bioactive molecules that are not required for growth but confer a fitness advantage to the producing organism. Certain pigments, including melanins, commonly require laccases (also called phenoloxidases) for synthesis. For instance, 1-8-dihydroxynapthalene (DHN), the melanin found in *A. fumigatus* spores, requires two copper-dependent phenoloxidases, Abr1 and Abr2, for the final step of its production [24]. Copper-dependent laccases also participate in the degradation of lignocellulose [25]. Several other enzymes use copper as a cofactor or contain copper-binding sites that have yet to be further investigated (Table 1). 

### 2.2. Copper Transporters and Copper-dependent Transcription Factors

Copper needs to be transported into the cell and delivered to copper-dependent enzymes. *A. fumigatus* possesses both high-affinity (CtrC and CtrA2) and one low-affinity copper importer (Ctr2), which are expressed to ensure copper sufficiency under conditions when copper is limited [28]. It is unknown whether the copper transporter CtrA1 is high affinity or low affinity. The copper-specific exporter CrpA is used to pump excess copper out of the cell [26]. The copper transporters are regulated by the copper-dependent transcription factors, AceA and MacA, which are also responsible for regulating other genes associated with copper homeostasis [26,27]. Details on copper transport and regulation in fungi will be described in Section 4 of this review.

## 3. Copper Toxicity

While it is an essential cofactor for many proteins, copper also has toxic characteristics. The element participates in Fenton chemistry, generating hydroxyl radicals by reacting with hydrogen peroxide, a natural by-product of aerobic respiration [33,34]. Hydroxyl radicals damage DNA by inducing strand breaks, oxidizing nucleoside bases and inactivating iron–sulfur cluster-containing enzymes, a process shared with hydrogen peroxide [35,36]. ROS inactivation of iron–sulfur cluster proteins in yeast has been demonstrated by Murakami and Yoshino by treating yeast with paraquat, an ROS generator. This resulted in the inactivation of aconitase, an enzyme containing an iron–sulfur cluster [37]. 

Copper can also cause mismetallation, where it replaces metal cofactors in proteins, rendering them inactive [38]. Proteins that depend on solvent-exposed iron–sulfur clusters for single-electron transfer, such as fumarase A and aconitase in the citric acid cycle, or dehydratases that are responsible for branched-chain amino acid biosynthesis, are particularly vulnerable [36]. The toxic effects of copper on iron–sulfur clusters has been investigated in the context of the Yah1 protein in yeast. Yah1, a mitochondrial ferredoxin containing an iron–sulfur cluster and required for electron transfer, is vulnerable to copper-mediated damage and when *YAH1* is overexpressed, it results in a strain with increased resistance to copper toxicity [39]. *A. fumigatus* contains several iron–sulfur cluster enzymes, but it is unknown if copper-mediated mismetallation damage exists in this fungus. Copper can specifically affect the homeostasis of iron since copper is needed for FetC, a ferroxidase that reduces iron prior to import [23]. Potentially implicating ROS generated by copper, a study by De Freitas et al. demonstrated that *S. cerevisiae* lacking the superoxide dismutase, *sod1*, resulted in an induction of genes involved in iron acquisition. They hypothesized that this is most likely due to the need for iron–sulfur cluster biogenesis to replace those that have been inactivated due to ROS [40]. 

## 4. Infection Biology and Copper 

During infection, pathogenic fungi typically encounter elevated toxic copper levels as a result of a protective host response. Macrophages are activated and express high levels of the copper transporter Ctr1p, raising intracellular copper levels. The Cu^+^-transporting P-type ATPase ATP7Ap is transported from the Golgi to the phagolysosomal membrane, pumping copper to toxic levels within the phagolysosome, killing the ingested pathogen [41,42]. Although host copper sequestration is not a widespread response to infection, it has been reported for *C. albicans* and *C. neoformans* infecting the kidneys and brain, respectively, suggesting that fungi may encounter copper starvation in specific body niches [43,44,45].

### 4.1. Fungal Response to Copper Limitation 

Copper homeostasis in fungi is mediated by the transcriptional regulation of genes involved in copper uptake, sequestration, and removal (Figure 1). In response to low cellular copper concentrations, fungal transcription factors regulate the expression of genes responsible for the uptake and absorption of copper. In *S. cerevisiae*, the nuclear-localized transcriptional regulator Mac1 is activated under low copper conditions. Two conserved cysteine-rich motifs in the carboxy-terminal of Mac1 bind up to eight Cu^+^ atoms, resulting in an inhibitory intramolecular interaction with the amino-terminal DNA-binding domain. Under low copper, the bound Cu^+^ atoms dissociate, the inhibitory interaction is released, and Mac1p binds conserved copper responsive elements (CuREs) in the promoters of target genes. These include the plasma membrane Cu/Fe reductase encoded by *FRE1* and the high-affinity Cu^+^ transporters encoded by *CTR1* and *CTR3* [46,47,48]. The cell-surface reductase Fre1 reduces exogenous Cu^+2^ to Cu^+^, which can then be transported into the cell by Ctr1 and Ctr3. These high-affinity Cu^+^ transporters contain three transmembrane domains that form a trimeric channel and an extracellular methionine- or cysteine-rich region that funnels Cu^+^ into the pore. Inside the yeast cell, excess Cu^+^ is sequestered by the metallothioneins Cup1 and Crs5, and by glutathione [49,50,51]. Cu^+^ is also directed to the desired cellular compartments by the dedicated Cu^+^ chaperones Atx1 and Ccs1, which respectively transport copper to the Ccc2p Cu^+^ transporter in the secretory compartment and the superoxide dismutase Sod1p that degrades oxygen radicals. An as yet unidentified chaperone directs Cu^+^ to the mitochondrial Cox17 chaperone that transfers it to the mitochondrial cytochrome C proteins Cox1 and Cox2, involved in the electron-transport chain [29,52,53].

As in yeast, *C. albicans* counters low copper levels by activating Mac1 to transcribe *FRE7* copper reductase and *CTR1* high-affinity Cu^+^ transporter [54] (Figure 1C). In contrast, *C. neoformans* expresses a single dual-function transcription factor, Cuf1, that controls the response to both low and high copper to induce expression of the high-affinity Cu^+^ transporters Ctr1/Ctr2 or the metallothioneins Cmt1/Cmt2, respectively [45,55,56] (Figure 1B). The effect of *MAC1* deletion on *C. albicans* virulence has not been tested. Deletion of *C. neoformans Cuf1* or overexpression of *Ctr1/Ctr2* results in increased lung fungal load in infected mice [45,55].

Within the genus *Aspergillus*, the response to copper limitation has only been characterized in *A. fumigatus* [26,28,57,58,59] (Figure 1D). The *A. fumigatus* genome contains three genes encoding the copper-binding transcription factors MacA, responding to copper limitation; AceA, to copper excess; and CufA, whose role remains unclear [26]. 

Deletion of *macA* in *A. fumigatus* leads to reduced growth and conidiation under low copper [26,57,58]. Intracellular Cu^+^ levels are strongly diminished. As a result, the activity of both SOD and laccase, which use Cu^+^ as a cofactor, is impaired, leading to reduced resistance to oxygen radicals and reduced conidial pigmentation [59]. *A. fumigatus* MacA binds conserved 5′-TGTGCTCA-3′ motifs in the promoters of the high-affinity Cu^+^ transporters CtrA1, CtrA2, and CtrC, leading to their transcriptional activation [26,57,58]. Interestingly, overexpression of CtrA2 or CtrC rescues the copper-sensitive phenotype of the *macA*-null mutant, while deletion of both transporters phenocopies the *macA*-null mutant. This implies that the principal targets regulated by MacA are CtrA2 and CtrC [28,58]. 

RNA-seq analysis reveals that the main cellular functions affected by *A. fumigatus macA* deletion are oxidation–reduction, metabolism, and transmembrane transport, including strong downregulation of the high-affinity Cu^+^ transporters CtrA1, CtrA2, and CtrC and differential expression of numerous metal, ATP-binding cassette (ABC), and major facilitator superfamily (MFS) transporters [58]. Park et al. [59] showed by microarray and northern blot analysis that following *macA* deletion, genes involved in iron siderophore synthesis (*sidA*, *sidD*), siderophore transport (*mirB*, *mirD*, and *sit1*), reductive iron transport (*ftrA*, *fetC*), and iron response regulator *hapX* are downregulated. ChIP-seq and EMSA were used to identify the MacA-binding motif in the promoter region of these genes. MacA localized to the nucleus under iron- or copper-depleted conditions and was mostly detected in the cytoplasm under iron- or copper-replete conditions [60]. Together, these results suggest that MacA may function as a bifunctional transcription factor of copper and iron metabolism in *A. fumigatus*, a situation that is unexpected and distinct from that found in other organisms. 

The involvement of MacA in *A. fumigatus* virulence is contradictory, despite the use of apparently identical strains and mouse models. Wiemann et al. [26] showed that the *macA*-null strain was unaltered in virulence and in susceptibility to killing by mouse alveolar macrophages. In contrast, Cai et al. [27] and Park et al. [61] showed attenuated virulence, reduced fungal load, and increased susceptibility to macrophage killing of the *macA*-null strain. These differences could be associated with the heterogeneity of fungal isolates [62].

### 4.2. Fungal Response to Copper Excess

In response to high extracellular copper concentrations, fungal transcription factors regulate the expression of genes that are responsible for sequestration and efflux of excess copper (Figure 1). In *S. cerevisiae*, the transcriptional regulator Ace1 is activated under high copper conditions. Binding of four Cu^+^ atoms to the single Ace1 cysteine-rich Cu-binding domain triggers intramolecular conformational changes within the adjacent amino-terminal DNA-binding domain. Ace1 then binds to conserved motifs in target genes that include *CUP1* and *CRS5*, encoding metallothioneins and *SOD1*, encoding superoxide dismutase [49,63,64,65,66]. Cup1, Crs5, and Sod1 suppress copper toxicity by sequestering the excess metal (Figure 1A). Interestingly, the protective role of SOD1 in copper buffering seems unrelated to its superoxide scavenging activity, as the enzyme protects against copper toxicity under anaerobic as well as aerobic conditions [67]. 

*C. albicans* and species of *Aspergillus* also sense elevated copper through Ace1 homologs. Ace1/AceA contains a conserved zinc-binding domain and (R/K)GRP ((Arg/Lys)-Gly-Arg-Pro) sequence motif essential for DNA minor groove site-specific binding and function, and eight cysteine-rich residues that form a polycopper cluster that binds four Cu^+^ ions cooperatively [58]. Unlike in *S. cerevisiae* and *C. neoformans*, the copper-buffering system *in C. albicans* and species of *Aspergillus* relies primarily on the AceA-dependent transcriptional activation of *CRP1*/*crpA* encoding a Cu^+^ P-type ATPase (Figure 1C,D). CRP1/CrpA actively transports excess copper from the cytoplasm to the extracellular environment [26,58,68,69,70]. CRP1/CrpA contains eight transmembrane domains, a conserved CPC (Cys-Pro-Cys) copper translocation motif in the sixth transmembrane segment and cysteine-rich metal-binding motifs in the cytoplasmic N-terminal and is apparently localized in the endoplasmic reticulum and plasma membrane [58,68,69]. *Aspergillus flavus*, which is both a plant and animal pathogen, contains two redundant AceA-activated *crpA* homologs, *crpA* and *crpB*. Deletion of both homologs is necessary for induction of copper sensitivity. The crpA/crpB-null mutant exhibits attenuated virulence in infected mice but retains full activity in infecting corn seeds [70]. Interestingly, in *S. cerevisiae*, the CRP1 homolog Ccc2 plays a completely different role in transporting Cu^+^ from the cytosolic chaperone protein Atx1 into the ER where it is incorporated into Fet3 Cu-oxidase required for reductive iron uptake. It is not known whether Crp1/CrpA also accepts Cu^+^ from a dedicated cytosolic chaperone protein similar to yeast Atx1 or whether they directly transport excess cytosolic Cu^+^. 

In *C. albicans*, Crp1 is responsible for the high resistance to copper, whereas the metallothionein Cup1 is responsible for the residual copper resistance [68]. *CRP1* deletion attenuates *C. albicans* virulence in infected mice, as assessed by kidney fungal load [44].

*A. fumigatus* deletion mutants of *aceA* and *crpA* are hypersensitive to both elevated extracellular copper and ROS in vitro, suggesting these two stresses are inextricably connected. Both strains accumulate higher copper levels and show greater susceptibility to killing by macrophages. In a mouse model of infection, these mutants display reduced growth and virulence [26,58]. Overexpression of CrpA in the *aceA*-null background reestablishes a wild-type phenotype, confirming that CrpA is the major effector target gene of AceA. Deletion of the single *A. fumigatus* metallothionein-encoding gene *cmtA* does not affect growth on elevated copper conditions or resistance to macrophage challenge [58,71]. However, overexpression of CmtA in the *crpA*-null background provides partial protection against high Cu^+^ indicating that metallothioneins may play a minor role in protecting this fungus against toxic levels of copper [58]. Interestingly, Cai et al. [58] also demonstrated increased sensitivity to zinc following deletion of *aceA* and *crpA*, but we were unable to replicate these results in our strains possibly due to differences in genetic backgrounds.

The *A. fumigatus* genome contains two additional putative Cu^+^-transporting P-type ATPase genes, *ctpA* and *pcaA* [24,31]. CtpA is important for conidial melanization under copper limitation, most likely by supplying the metal to the conidial laccases Abr1/2 [24]. PcaA is not a copper transporter and has been recently shown to provide protection against cadmium, apparently by efflux of this toxic metal [31].

## 5. Copper-Binding Secondary Metabolites

As described above, fungi contain several copper transporters that import Cu^+2^ from the environment [26,27]. This process can be likened to iron import, where FtrA transports Fe^+3^ into the cell [23]. However, many fungi (and bacteria) also synthesize specialized secondary metabolites that acquire iron from the environment; these Fe chelating metabolites are commonly known as siderophores [72]. Unlike other human pathogenic fungi, *A. fumigatus* contains over 30 biosynthetic gene clusters (BGC) that produce secondary metabolites [73]. Two of these BGCs encode for the production of siderophores, the extracellular siderophore triacetylfusarinine, shown to be critical for iron uptake and virulence in murine models [23] and the intracellular siderophore ferricrocin [74]. A third BGC encodes for an iron-binding metabolite, hexadehydroastechrome, which is important for iron homeostasis and enhances virulence in murine models of IA when overproduced [71,75]. 

Like fungi, bacterial utilize siderophores for iron uptake and iron homeostasis but bacteria also have been found to synthesize small molecules for analogous functions in copper biology. Chalkophores, the copper analog of siderophores, are used for copper uptake and as a mechanism to mitigate copper-mediated damage in bacteria [76]. For example, coproporphyrin III is used by the denitrifying bacteria *Paracoccus denitrificans* to ensure adequate copper supply for the denitrifying process (Figure 2B) [20,77]. Supplementing copper-depleted growth media with coproporphyrin III remediates the copper growth defect of *P. denitrificans* suggesting a role for coproporphyrin III in copper uptake [78]. Methanobactin is a chalkophore produced by methylotrophic bacteria and is secreted to acquire copper for methane monooxygenase, an enzyme used to convert methane to methanol (Figure 2A). Yersiniabactin, originally described as a siderophore and associated with the virulence of members of the Enterobacteriaceae, has recently been shown to bind copper in addition to iron [79]. Interestingly, the yersiniabactin metallophore system has been implicated in both detoxification of excess copper and the uptake of copper (Figure 2D) [80]. Isolates that produce yersiniabactin are more resistant to copper toxicity suggesting a role for its protective effect. In addition, isolates that do not produce yersiniabactin but are supplemented with purified yersiniabactin regain resistance to toxic levels of copper [81]. Yersiniabactin has also been shown to increase the bioavailability of copper, leading to an adequate supply for functionality of the amine oxidase, TynA [82]. Another chalkophore is the *Streptomyces thioluteus* isocyanide compound SF2768 that has been shown to chelate Cu^+^ and is important for copper uptake (Figure 2C) [83]. 

Isocyanides in particular have metal-chelating capabilities including copper [84], and certain bacteria produce isocyanides to promote metal homeostasis or inhibit copper-dependent enzymes [85,86]. Recently, isocyanide synthase BGCs have been found in *A. fumigatus*, and one of these BGCs was shown to synthesize a series of isocyanides including xanthocillin (Figure 2E) [87]. Deletion of the isocyanide synthase XanB eliminated xanthocillin production in *A. fumigatus*. Additionally, a second isocyanide synthase-containing protein, CrmA, was located in another BGC termed the *crm* (copper responsive metabolite) BGC although the encoded metabolite was not characterized. The *crm* cluster genes are upregulated during copper starvation and the *xan* cluster genes during copper excess [87]. Furthermore, MacA positively regulates the *crm* genes whereas AceA positively regulates *xan* gene expression, thus firmly tying regulation of both metabolites with the copper regulon. Current studies are aimed at determining whether *A. fumigatus* isocyanides are important in the copper homeostasis and virulence of this organism. 

## 6. Future Directions

There are several gaps in the knowledge of copper homeostasis in *A. fumigatus* and how the host immune system manipulates the metal in response to infection. Eukaryotes need to carefully manage the import, compartmentalization, delivery, and export of copper. For most fungi, including *A. fumigatus*, global knowledge of all the triggers that induce or repress the copper regulon is incomplete, and it is unknown whether small molecules, such as fungal isocyanides, play a role in copper homeostasis. Intracellular copper storage and chaperoning as well as several putative copper-binding proteins have not yet been fully investigated in *A. fumigatus*. Gaining a better understanding of copper homeostasis in this opportunistic human pathogen has the potential to lead to a better understanding of how *A. fumigatus* is able to manage copper availability in the environment as well as provide novel treatment or prophylactic insights in the context of invasive aspergillosis.

## Figures and Tables

**Figure 1 ijms-20-01980-f001:**
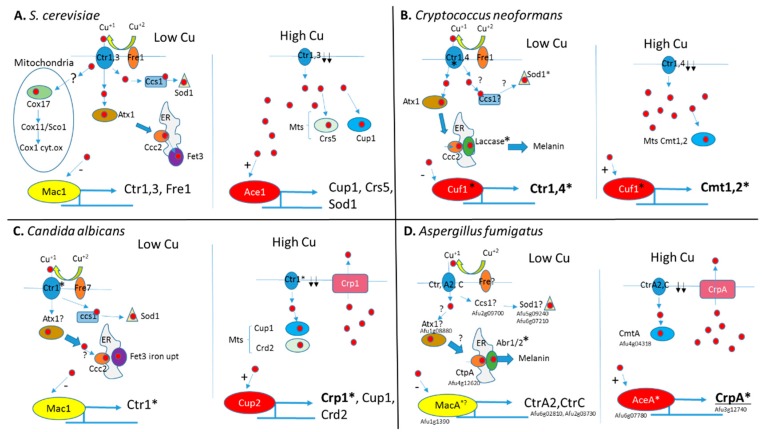
Schematic representation of copper homeostasis in *Saccharomyces cerevisiae*, *Cryptococcus neoformans*, *Candida albicans*, and *A. fumigatus*. (**A**) In *S. cerevisiae*, under low Cu^+^, Fre1 reduces Cu^+2^ to Cu^+^ (red circles) for uptake by transporters Ctr1 and Ctr3. Inside the cell, Cu^+^ is bound by chaperone proteins Cox17, Atx1, and Ccs1 that transport Cu^+^ to the mitochondrial cytochrome oxidase Cox1, ER-localized Fet3 ferric reductase, and Sod1 superoxide dismutase, respectively. Low intracellular Cu^+^ levels are sensed by transcription factor Mac1 to activate genes encoding the copper transporters Ctr1, Ctr3, and Fre1 reductase. High intracellular Cu^+^ levels are sensed by Ace1 transcription factor to activate the genes encoding metallothioneins (Mts) Cup1 and Crs5 and superoxide dismutase Sod1. Cup1 and Crs5 bind excess intracellular Cu^+^ and Sod1 oxidizes oxygen radicals formed under excess Cu^+^. High intracellular Cu^+^ levels also inhibit Mac1 activation to downregulate *Ctr1* and *Ctr3* expression. (**B**) In *C. neoformans* under low Cu^+^, Fre1 reduces Cu^+2^ to Cu^+^ (red circles) for uptake by transporters Ctr1 and Ctr4. Inside the cell, Cu^+^ is bound by chaperone proteins Atx1 and possibly a Ccs1 homolog. Atx1 transports Cu^+^ to the ER-localized laccase involved in melanin biosynthesis. A Ccs1 homolog is predicted to transport Cu^+^ to Sod1 superoxide dismutase. Low intracellular Cu^+^ levels are sensed by the transcription factor Cuf1 to activate the genes encoding the copper transporters Ctr1 and Ctr4. High intracellular Cu^+^ levels are also sensed by Cuf1 to activate the genes encoding metallothioneins (Mts) Cmt1 and Cmt2 to bind excess Cu^+^ and downregulate copper transporters Ctr1 and Ctr4. (**C**) In *C. albicans* under low Cu^+^, ferric reductase Fre7 reduces Cu^+2^ to Cu^+^ (red circles) for uptake by Cu^+^ transporter Ctr1. Inside the cell, Cu^+^ is bound by chaperone protein ccs1 that provides Cu^+^ to superoxide dismutase sod1. A putative Atx1 homolog is proposed to transfer Cu^+^ to the ER-Cu^+^ transporter Ccc2, providing copper for the Fet3 ferric reductase involved in iron uptake. Low intracellular Cu^+^ levels are sensed by transcription factor Mac1 to activate Ctr1 encoding copper transporters. High intracellular Cu^+^ levels are sensed by transcription factor Cup2 to activate the genes encoding Crp1 copper exporter, Cup1 and Crd2 metallothioneins to respectively remove or bind excess Cu^+^. (**D**) In *A. fumigatus* under low Cu^+^, an unknown ferric reductase (Fre?) reduces Cu^+2^ to Cu^+^ (red circles) for uptake by transporters CtrA2 and CtrC. Inside the cell, Cu^+^ presumably binds uncharacterized chaperone proteins homologous to yeast Atx1 and Ccs1. The ER-Cu^+^ transporter CtpA provides copper for the conidial laccases Abr1 and Abr2 that generate melanin. Low intracellular Cu^+^ levels are sensed by transcription factor MacA to activate genes encoding the copper transporters CtrA2 and CtrC. High intracellular Cu^+^ levels are sensed by AceA to activate CrpA encoding a copper exporter. Induced overexpression of the metallothionein CmtA also partially protects against Cu^+^ excess. *A. fumigatus* gene designations are provided. Genes whose deletion reduces virulence are marked by an asterisk *.

**Figure 2 ijms-20-01980-f002:**
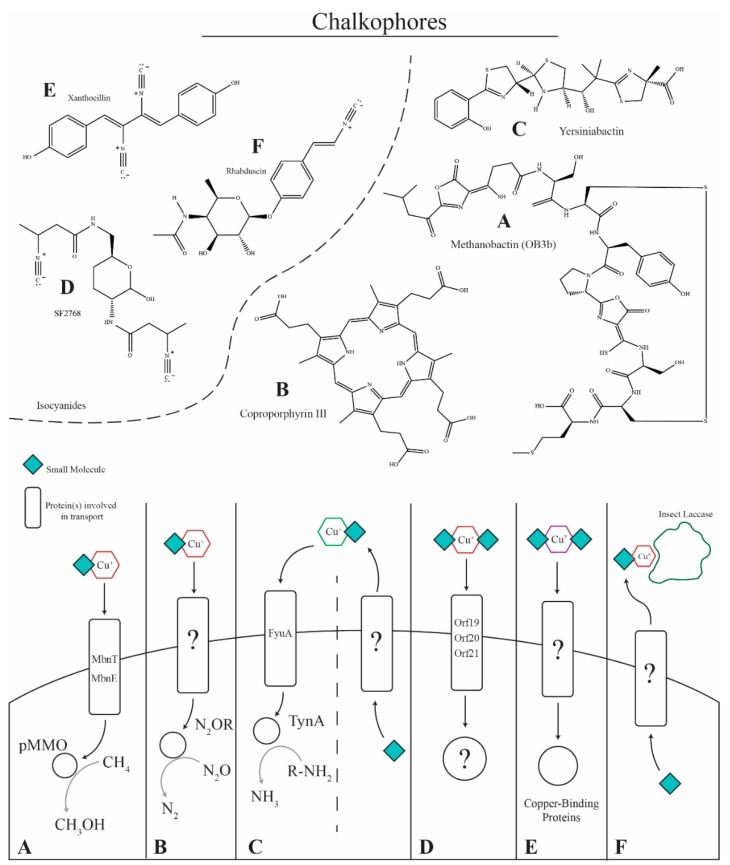
Copper-binding small molecules. (**A**) Methane-oxidizing bacteria such as *Methylosinus trichosporium* OB3b produce methanobactins to acquire copper for the particulate methane monooxygenase (pMMO) enzyme. Methanobactin is secreted via an unidentified mechanism, binds copper in the environment, and is transported into the cell via MbnT and MbnE for delivery to pMMO for the oxidation of methane to methanol. (**B**) *Paracoccus denitrificans* produces the chalkophore coproporphyrin III that is transported via an unknown mechanism to bind copper in the environment and deliver it into the cell for use in nitrous oxide reductase. (**C**) Yersiniabactin is produced by bacteria containing the *Yersinia* high pathogenicity island (HPI) and is implicated as a virulence factor for enteropathogenic *Escherichia coli*, binding copper and preventing it from damaging the pathogen. The compound also acts as a chalkophore, being required for copper sufficiency, binding copper in the environment and transporting it into the cell for the amine oxidase, TynA. (**D***) Streptomyces thioluteus* has been shown to produce the isocyanide chalkophore SF2768 that is required for copper uptake. The chalkophore is produced and transported via putative transporter proteins, binds copper, and transports it back into the cell. (**E**) The isocyanide xanthocillin and xanthocillin-like derivatives produced by *A. fumigatus* are proposed to act as chalkophores, where they are secreted by an unknown mechanism, bind copper in the environment, and transport it back into the cell for use in copper-dependent enzymes such as cytochrome c oxidase, nitrite reductase, amine oxidase(s), superoxide dismutase (SOD1), and laccases (Abr1/Abr2). (**F**) The insect pathogen *Xenorhabdus nematophila* produces the isocyanide, virulence factor rhabduscin, which inhibits the copper-dependent insect laccase. The laccase is essential for producing melanin, a component of the insect immune response.

**Table 1 ijms-20-01980-t001:** Proteins that bind copper or have predicted copper-binding sites. Proteins that contain ‘*’ are named after homologous genes that have been characterized in other fungi.

Copper-Binding Proteins in *Aspergillus fumigatus*
	Designation (AFUA)	Name	Function	Reference
Known function	6g07780	AceA	Copper-Toxicity Transcription Factor	[26]
1g13190	MacA	Copper-Deficiency Transcription Factor	[27]
2g01190	CufA	Unknown Function Transcription Factor	[26]
6g02810	CtrA2	High-Affinity Copper Importer	[28]
2g03730	CtrC	High-Affinity Copper Importer	[28]
3G13660	CtrA1	Copper Importer	[28]
3g08180	Ctr2	Low-Affinity Importer	[28]
3g12740	CrpA	Copper P-Type ATPase Exporter	[26]
4g12620	CtpA	Intracellular Copper ATPase	[24]
4g04318	CmtA	Copper Metallothionein	[26]
5g09240	Sod1	Cytoplasmic Superoxide Dismutase	[19]
5g03790	FetC	Ferrioxidase Involved in Iron Import	[23]
2g17540	Abr1	Laccase	[24]
2g17530	Abr2	Laccase	[24]
2g13110	CycA	Cytochrome C oxidase	[14]
Putative function	3g14950	NirK *	Copper-Binding Nitrite with Reductase Activity	[20]
1g08880	Atx1 *	Copper Chaperone	[29]
2g09700	Ccs1 *	Sod1 Copper Chaperone	[30]
1g16130	Pca1 *	Cd-Exporting ATPase	[31]
3g07690	Cox17 *	Copper Chaperone to Cytochrome C Oxidase	[32]
5g01470		Amine Oxidase	
3g14590		Amine Oxidase	
3g00680		Copper Amine Oxidase	
7g08470		Copper Amine Oxidase	
5g07360		Peroxisomal Amine Oxidase

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
