# Peer review of "Copper Utilization, Regulation, and Acquisition by *Aspergillus fumigatus"

_ijms, 2019, doi:10.3390/ijms20081980_

Round 1

Reviewer 1 Report

Manuscript ID: ijms-476889

Title: Copper Utilization, Regulation, and Acquisition by Aspergillus fumigatus

Comments

This is a timely and excellent review of copper homeostasis in Aspergillus fumigatus.  It is well written and provides an up to date analysis of the current state of knowledge.  The review raises questions that remain to be addressed in the field, especially in pathogenic fungi.

Minor issues are suggested/indicated below.

Line 46.       The authors ground their review in studies on A. fumigatus.  They also refer to studies in S.      cerevisiae, C. neoformans, and C. albicans.  Unfortunately, they have not mentioned      S. pombe from which novel copper-binding proteins have been identified and      characterized, including copper amine oxidase and cell-cycle-specific      copper transporter.

Line 53.       …"various" copper chaperones.  The number of proven copper chaperones      is very low in comparison with the number of copper-containing proteins      (which themselves are relatively few in number in unicellular and      multicellular organisms).  The      authors may provide a better description by specifying that only few      copper chaperones have been shown to be responsible for delivery of copper      to a specific copper-dependent protein.

Line 82. Typo: Abr1.

Line 96. Typo: section 4 instead of      section IV.

Line 104. Typo: paraquat.

Line 117. Typo: et.,

Line 132.       Genes encoding proteins involved in delivery of copper to proteins      (e.g. copper chaperones) are generally not regulated at the      transcriptional level as a function of copper availability.  Thus, "intracellular      transport" should be removed of the sentence.

Line 144.       "free" should be replaced by "excess".  Few times in the text of the manuscript,      "free" copper is used.  It      is surprising since it has been shown that there is virtually no free      copper in the cell by the O’Halloran Group.

Line 146.       This sentence is unclear since in the case of Cox17, results have      shown that Cox17 localized exclusively to the mitochondria is sufficient      for delivery of copper to cytochrome c oxidase.  This suggests that either an as yet      unidentified chaperone or a small molecule carrier is responsible for      trafficking of copper from the plasma membrane to Cox17 in the      mitochondria.

 Lines      199 – 200.  The authors should      insist on the fact that copper chaperones become critical for cell      function only under copper limitation conditions.  In contrast, metallothioneins are      primarily required in response to high concentrations of copper.  These two cellular copper conditions (poor      versus replete-conditions) should be illustrated separately in Fig. 1,      otherwise the reader would interpret as a choice for copper ions, either      copper chaperone or metallothionein, which is not the case.

 In      C. neoformans, it is still unclear whether Cuf1 directly binds a cis-regulatory      element upstream of MT1, MT2, and ATM1 genes.  Panel B of Fig. 1 should be modified      under high copper conditions.

Line 210. Typo: a copper transporter.

Lines 212-213.  In A. fumigatus, is it proven that      copper ions bind the CmtA protein?

 Line      226. "free" should be replaced by "excess". 

Line 232.       What means "in yeast" ?       S. cerevisiae?  Caveat.  S. cerevisiae is quite often different      as compared with other fungal species.       

Author Response

Please find uploaded our revised manuscript ID ijms-476889 entitled “Copper Utilization, Regulation, and Acquisition by Aspergillus fumigatus”.  We thank the reviewers for their comments and have addressed all of the minor revisions they suggested.  Our replies are in bold.

Sincerely,

Nancy P. Keller

Reviewer #1

Comments

This is a timely and excellent review of copper homeostasis in Aspergillus fumigatus. It is

well written and provides an up to date analysis of the current state of knowledge. The review

raises questions that remain to be addressed in the field, especially in pathogenic fungi.

Reply- thank you for your kind words.

Minor issues are suggested/indicated below.

1. Line 46. The authors ground their review in studies on A. fumigatus. They also refer

to studies in S. cerevisiae, C. neoformans, and C. albicans. Unfortunately, they have

not mentioned S. pombe from which novel copper-binding proteins have been

identified and characterized, including copper amine oxidase and cell-cycle-specific

copper transporter.

Reply- A reference to S. pombe has now been included.

2. Line 53. …"various" copper chaperones. The number of proven copper chaperones is

very low in comparison with the number of copper-containing proteins (which

themselves are relatively few in number in unicellular and multicellular organisms).

The authors may provide a better description by specifying that only few copper

chaperones have been shown to be responsible for delivery of copper to a specific

copper-dependent protein.

Reply- The statement has been clarified, thank you.

3. Line 82. Typo: Abr1.

Reply- Corrected

4. Line 96. Typo: section 4 instead of section IV.

Reply- Corrected

5. Line 104. Typo: paraquat.

Reply- Corrected

6. Line 117. Typo: et.,

Reply- Corrected

7. Line 132. Genes encoding proteins involved in delivery of copper to proteins (e.g.

copper chaperones) are generally not regulated at the transcriptional level as a function

of copper availability. Thus, "intracellular transport" should be removed of the

sentence.

Reply- "intracellular transport" has been removed

8. Line 144. "free" should be replaced by "excess". Few times in the text of the

manuscript, "free" copper is used. It is surprising since it has been shown that there is

virtually no free copper in the cell by the O’Halloran Group.

Reply-"free" has been replaced by "excess" here and in lines 226 and 246.

9. Line 146. This sentence is unclear since in the case of Cox17, results have shown that

Cox17 localized exclusively to the mitochondria is sufficient for delivery of copper to

cytochrome c oxidase. This suggests that either an as yet unidentified chaperone or a

small molecule carrier is responsible for trafficking of copper from the plasma

membrane to Cox17 in the mitochondria.

Reply- Thank you! This has been corrected on lines 147-8

10. Lines 199 – 200. The authors should insist on the fact that copper chaperones become

critical for cell function only under copper limitation conditions. In contrast,

metallothioneins are primarily required in response to high concentrations of copper.

These two cellular copper conditions (poor versus replete-conditions) should be

illustrated separately in Fig. 1, otherwise the reader would interpret as a choice for

copper ions, either copper chaperone or metallothionein, which is not the case.

Reply-We have separated the two conditions (low, replete Cu) in a revised figure and revised figure legend.

11. In C. neoformans, it is still unclear whether Cuf1 directly binds a cis-regulatory

element upstream of MT1, MT2, and ATM1 genes. Panel B of Fig. 1 should be

modified under high copper conditions.

Reply-Recently Garcia-Santamarina (mol.micro 108 473 2018) used CHIP-seq and RNA seq analysis to define the Cuf1 regulon. It includes MT1, MT2 and ATM1.

12. Line 210. Typo: a copper transporter.

Reply-corrected

13. Lines 212-213. In A. fumigatus, is it proven that copper ions bind the CmtA protein?

Reply- No, although it was shown that overexpression of CmtA in the crpA null background provides partial protection against high Cu+. The sentence has been modified to “is apparently bound”.

14. Line 226. "free" should be replaced by "excess".

Reply-corrected.

15. Line 232. What means "in yeast" ? S. cerevisiae? Caveat. S. cerevisiae is quite often

different as compared with other fungal species.

Reply- yeast has been corrected to S. cerevisiae.

Reviewer 2 Report

The manuscript entitled "Copper Utilization, Regulation, and Acquisition by Aspergillus fumigatus" has been reviewed. The manuscript is organized and written very well. But I feel a few questions from it.

1. In fact, the copper metabolism in A. fumigatus is not studied well so far, and it is little bit difficult to write review paper about this topic. So the author cited many research paper of S. cerevisiae. For example, in section 2.1, the author explained about the relationship between iron and copper in iron uptake system. However, we do not know the machanism exactly in A. fumigatus. 

2. In section 2.2, actually Ctr1A does not complement yeast Ctr1, so I do not know whether Ctr1A is high affinity or not. The author should correct.

3. In section 4.1, from the virulence assay using murine model, there is no effect of MacA on virulence from the reference of 27. But if you read reference 27 carefully, you can find MacA affect to virulence in murine model. The authors of reference 27 corrected their data.

4. In section 4.2, CptA is CtpA.

5. Generally, MacA is not a correct name. Many paper call it as AfMac1. So I recommend the name of protein as AfMac1.

As I mentioned above, there are few research paper about copper metabolism in A. fumigatus. However, this review will help to understand copper metabolism in fungal system.

Author Response

Please find uploaded our revised manuscript ID ijms-476889 entitled “Copper Utilization, Regulation, and Acquisition by Aspergillus fumigatus”.  We thank the reviewers for their comments and have addressed all of the minor revisions they suggested.  Our replies are in bold.

Sincerely,

Nancy P. Keller

Reviewer #2

The manuscript entitled "Copper Utilization, Regulation, and Acquisition by Aspergillus fumigatus" has been reviewed. The manuscript is organized and written very well. But I feel a few questions from it.

1. In fact, the copper metabolism in A. fumigatus is not studied well so far, and it is little bit difficult to write review paper about this topic. So the author cited many research paper of S. cerevisiae. For example, in section 2.1, the author explained about the relationship between iron and copper in iron uptake system. However, we do not know the mechanism exactly in A. fumigatus.

Reply- This is a good point.  We have modified the statement for clarification.

2. In section 2.2, actually Ctr1A does not complement yeast Ctr1, so I do not know whether Ctr1A is high affinity or not. The author should correct.

Reply- Corrected in both Table 1 and the text.

3. In section 4.1, from the virulence assay using murine model, there is no effect of MacA on virulence from the reference of 27. But if you read reference 27 carefully, you can find MacA affect to virulence in murine model. The authors of reference 27 corrected their data.

Reply- Ref 27 (our work) shows no difference in virulence between WT and MacA null in terms of mortality or fungal load (Fig. 2). Later figures focused on AceA.

4. In section 4.2, CptA is CtpA.

Reply- this has been corrected in the entire manuscript.

5. Generally, MacA is not a correct name. Many paper call it as AfMac1. So I recommend the name of protein as AfMac1.

Reply- MacA was the term used in our publication (27) and is standard nomenclature for Aspergilli. Park et al and Cai et al used the term AfMac1, which is not standard.

As I mentioned above, there are few research paper about copper metabolism in A. fumigatus. However, this review will help to understand copper metabolism in fungal system.

Reply- Thank you